# Bone Tissue Engineering in the Treatment of Bone Defects

**DOI:** 10.3390/ph15070879

**Published:** 2022-07-17

**Authors:** Nannan Xue, Xiaofeng Ding, Rizhong Huang, Ruihan Jiang, Heyan Huang, Xin Pan, Wen Min, Jun Chen, Jin-Ao Duan, Pei Liu, Yiwei Wang

**Affiliations:** 1Jiangsu Provincial Engineering Research Center of Traditional Chinese Medicine External Medication Development and Application, Nanjing University of Chinese Medicine, Nanjing 210023, China; 20213108@njucm.edu.cn (N.X.); dingxiaofeng1993@163.com (X.D.); 20210615@njucm.edu.cn (R.H.); 20210618@njucm.edu.cn (R.J.); 20211615@njucm.edu.cn (H.H.); wenge1977@126.com (W.M.); chenjun75@163.com (J.C.); 2Jiangsu Collaborative Innovation Center of Chinese Medicinal Resources Industrialization, National and Local Collaborative Engineering Center of Chinese Medicinal Resources Industrialization and Formulae Innovative Medicine, Jiangsu Key Laboratory for High Technology Research of TCM Formulae, Nanjing University of Chinese Medicine, Nanjing 210023, China; 20193086@njucm.edu.cn (X.P.); dja@njucm.edu.cn (J.-A.D.); 3Burns Injury and Reconstructive Surgery Research, ANZAC Research Institute, University of Sydney, Concord Repatriation General Hospital, Concord 2137, Australia

**Keywords:** bone defect, tissue regeneration, 3D printing, scaffolds, biomaterials, tissue engineering

## Abstract

Bones play an important role in maintaining exercise and protecting organs. Bone defect, as a common orthopedic disease in clinics, can cause tremendous damage with long treatment cycles. Therefore, the treatment of bone defect remains as one of the main challenges in clinical practice. Today, with increased incidence of bone disease in the aging population, demand for bone repair material is high. At present, the method of clinical treatment for bone defects including non-invasive therapy and invasive therapy. Surgical treatment is the most effective way to treat bone defects, such as using bone grafts, Masquelet technique, Ilizarov technique etc. In recent years, the rapid development of tissue engineering technology provides a new treatment strategy for bone repair. This review paper introduces the current situation and challenges of clinical treatment of bone defect repair in detail. The advantages and disadvantages of bone tissue engineering scaffolds are comprehensively discussed from the aspect of material, preparation technology, and function of bone tissue engineering scaffolds. This paper also summarizes the 3D printing technology based on computer technology, aiming at designing personalized artificial scaffolds that can accurately fit bone defects.

## 1. Introduction

Hard bone constitutes approximately 15% of the total body weight and is known to be the largest organ system in the human body [1]. Bone tissue has a double-layered structure: The outer layer is cortical bone, which accounts for approximately 80% of the total adult bone mass and has a relatively dense porosity of 3–5%. It has high resistance to bending and torsion, and it is essential for physical support, structural integrity, and weight bearing. The cancellous bone formed by the inner layer of honeycomb-shaped trabecular connection accounts for about 20% of the total bone mass in adults with a high porosity of approximately 80–90% [2]. As an internal supporting system, the bone forms the skeleton of a human body and also plays an important role in maintaining motor function, hematopoietic function, and protecting the internal organs and nervous system [3].

Bone can regulate the metabolic requirements through calciotropic hormones (vitamin D3, parathyroid hormone, and calcitonin). In addition, better bone quality can improve the structure strength to better prevent fracture damage [4]. Contrary to societal misconceptions, bone tissue responds positively to high rates or frequency of stimulation [5]. A study found that regular walking did not significantly preserve the mineral density of the spinal bone in postmenopausal women [6]. In contrast, Watson et al. demonstrated that high-intensity impact training can preserve bone mass in postmenopausal women than low-intensity training [7]. Therefore, available data strongly suggest that exercise characterized by impact load is able to promote and maintain a person’s bone health throughout life.

Bone regeneration is the process of re-forming bone tissue with essential shape and function post partial bone tissue loss [8]. Such defects are caused by trauma, infection, tumor, or functional atrophy, and it is known as one of the most common injuries in clinical practice. Bone tissue can be self-repaired and regenerated, therefore, small defects normally heal without additional treatment [9]. However, when bone defect exceeds the critical size threshold (approximately > 2 cm) or greater than 50% loss of circumference of bone [10], it will cause nonunion, malunion, or pathological fracture [11]. According to the latest data, bone transplantation is second to blood transfusion in the world, and it is the second most common tissue transplantation [12]. Globally, 4 million people require bone transplantation or bone replacement surgery each year, while in the United States, the number of age-related bone disorders is expected to increase from 2.1 million in 2005 to 3 million in 2025 [13]. In Europe, the fracture cases raised about 28% from 2010 to 2025, with increased population [14]. Therefore, effective treatment and therapeutical of bone diseases has great clinical significance.

In this review, we overviewed bone tissue engineering scaffolds, including the current clinical treatment status, challenges, and future prospects. Moreover, advantages and disadvantages of various functional materials including organic materials, inorganic materials, and biological composite materials in bone tissue engineering were discussed, and the developmental direction of bone tissue engineering scaffolds was prospected.

## 2. Advances and Challenges of Bone Defect Treatment

### 2.1. Clinical Treatment

Clinical treatment of bone defects includes non-invasive and invasive therapies. Non-invasive therapeutical methods mainly refer to biophysical stimulation, pulsed electromagnetic field (PEMF), etc. Clinically used exogenous stimulation therapy, including electromagnetic field therapy, low-intensity ultrasound therapy, and hyperthermia can stimulate bone tissue re-growth with faster bone repair and minimal pain. Surgical treatment is the most commonly used method for reconstructing bone defects, and the most important treatment method includes bone graft, which refers to the graft from the common site to the recipient site, accounting for about a quarter of the surgical treatment of bone defects. Prosthetic surgery can be used to reconstruct or improve the process of defective, damaged, or lost structures. In addition, surgical treatment methods for bone defect repair include: Ilizarov technique, Masquelet technique, Arthroplasty, replacement, etc. (Figure 1).

#### 2.1.1. Bone Grafting

Bone grafting is widely used in clinical practice. It can be sub-grouped to autologous bone grafting and allogeneic bone grafting. Autologous bone has dual effects of osteo-induction and osteo-conduction, and it is the “gold standard” for bone repair with proved osteogenesis [15]. However, because of a lack of donor tissue together with additional secondary defects, it is not appropriate for children or elderly patients. Allogeneic and xenografting bone grafting are limited in use due to their insufficient integration and vascularization in the host area and potential risks of immune rejection and pathogen transmission. Cryogenic treatment can reduce the occurrence of immune rejection, but its mechanical strength and bone-induced activity is known to be correspondingly reduced.

#### 2.1.2. Ilizarov Technique

Ilizarov technique was proposed and named after a former soviet doctor in 1951 [16]. The core biomechanical theory of this technique is the “law of tension stress” (LTS), which states that continuous slow traction stimulation can promote tissue regeneration and active growth of biological tissue similar to embryonic tissue [17]. This rule is referred to as distraction osteogenesis (DO) in orthopedics, which Dr. Ilizarov first described in a canine model. Despite advantages of Ilizarov technology, it is undeniable that the technology also has disadvantages, such as complex operations and extremely long treatment and recovery [18]. In comparison with traditional methods, Ilizarov technique shows better outcomes in the treatment of fracture complicated with infection and large bone defects [19].

#### 2.1.3. Masquelet Technique

Membrane induction regeneration technique (Masquelet technique) is effective for the treatment of bone defects. Masquelet technique has two steps: Firstly, bone cement is filled in the defect area, aiming to induce the formation of local pseudoperiostium. Thereafter, cement filler is removed and replaced with autologous or allogeneic cancellous bone [20]. This technique can restore functional activity of bone tissue in a short time. However, due to the high cost, secondary surgery, and the limitation of bone graft volume, it cannot meet all requirements for large-segment bone defects.

#### 2.1.4. Bone Graft Substitutes

Bone graft substitutes can be made from synthetic or natural biomaterials. Various biomaterials are currently under developed or studied as bone graft scaffolds, including collagen, hydroxyapatite (HA), β-tricalcium phosphate (β-TCP), calcium phosphate cement, and ceramic glass. At present, the reconstruction of large bone defects is the goal in clinical practice, while titanium alloy is commonly used. Titanium is non-toxic, harmless, and has good corrosion resistance, and its elastic modulus is closer to human hard tissue. However, the surface of titanium alloy is relatively smooth resulting in poor osseointegration performance [21]. The introduction of 3D-printed titanium alloy can solve this problem via precisely controlled construction that mimics natural bone tissue at both macro and micro levels. Porous titanium alloy structure is also utilized to promote the adhesion of osteoblasts, providing enough space for the growth of bone tissue and promoting the growth of new bone tissue into the gap [22].

### 2.2. Challenges of Bone Defect Treatment

#### 2.2.1. Angiogenesis and Vascularization

The integrity of bone, as a highly vascularized tissue, relies on angiogenesis and the tight connection of bone cells in time and space. Therefore, sufficient angiogenesis plays a key role in bone development and repair [23]. Local vascularization in the early stages post grafting provides various essential nutrients for osteogenesis activity, but also plays an indispensable role between bone and adjacent tissues and organs [24]. In adults, the vascular endothelium is usually in a state of silence due to cell-to-cell contact and inhibition of cell proliferation. Only under certain stimulation, endothelial cells can be triggered for angiogenesis, such as: hypoxia environment, cell morphological changes, and changes in the sensitivity of angiogenic factors [25]. At present, common tissue engineering scaffolds are still greatly restricted in terms of material transfer exchange and neovascularization [25,26]. Findings showed that angiogenesis is mainly concentrated around the surface of scaffolds, leading to low nutrients and oxygen transfer to the internal area of the scaffold. Such insufficient oxygen supply can result in death of cells and necrotic-tissue due to hypoxia, hindering the growth of host bone cells, and cavitation or necrosis inside the scaffold [27].

#### 2.2.2. Osteoinduction and Osteoconduction

Bone induction refers to the process of stem cell differentiation into osteoblast cell lines under stimulation via the microenvironment. Bone induction process directly determines the success or failure of the bone regeneration process. Bone scaffolds can promote stem cell-directed differentiation by producing differentiation-inducing substances in the microenvironment, mainly ions and scaffolds. The ions that are precipitated from the scaffold material itself [28], and the scaffold as a delivery carrier, transporting growth factors or drugs, can induce stem cell differentiation. Bone morphogenetic proteins and vascular endothelial growth factor have also been found to promote bone induction in scaffolds [29,30].

Osteoconduction is the frame structure required for the growth of bone cells, and it provides a track for integration and migration of bone tissue. Bone tissue, capillaries, and surrounding tissue can gradually grow into the pores and form new bone tissue. Osteo-conduction is highly dependent on tissue interaction with biomaterials. This process involves cellular behaviors such as cell adhesion, attachment, proliferation, and migration. Scaffolds properties in terms of physicochemical strength and structural characteristics effect the osteoconductivity of the scaffolds.

#### 2.2.3. Osseointegration

The concept of bone integration was proposed by Brånemark, indicating direct connection of the non-fibrous connective tissue interface layer between implanted scaffold materials and the bone tissue [31]. Osteoblasts are allowed to form clusters on the surface of implants with an extracellular matrix, initiating the formation of new bone. Due to bone conversion and repair process, osteogenesis occurs at all stages of life. Thus, osseointegration can be described as the final step in the healing process of bone surrounding the implant. To promote osseointegration, modification of prosthesis materials surface via sand spraying, acid etching, laser treatment [32,33], together with alteration of bone metabolism around the prosthesis, can be both utilized.

## 3. Tissue Engineering Technologies in Bone Regeneration and Repair

The concept of “tissue engineering” was firstly introduced in 1996, while bone tissue engineering has become the most rapidly developed research field in tissue engineering [34]. Bone tissue engineering uses scaffolds, well-integrated cells, and bioactive growth factors to promote bone repair and regeneration, providing an innovative platform for regenerative medicine. The hydrophilicity, hydrophobicity, and biochemical structure of scaffold can affect cell adhesion, arginine–glycine–aspartate (RGD), and varying biological domains on scaffold surface are known to improve cell adhesion [35]. In addition, literature on different biomaterials and cell sources indicates a wide range of average pore sizes from 20 to 1500 µm for optimal cell attachment and successful bone regeneration [36,37,38]. Other studies have shown that pores ≥ 300 µm are critical for inducing direct osteogenesis and allow higher cell infiltration, migration, capillaries, and bone ingrowth [38,39,40]. The pore size and porosity of the scaffold determines total surface area in supporting tissue regeneration, but if the pore size is below 100 µm, cell migration and nutrient diffusion will be limited, resulting in a dead cavity in the internal area of the scaffolds [41]. In contrast, when pore size is over 750 µm, the specific surface area of the scaffold will decrease, and the mechanical properties of the scaffold will be reduced [42].

### 3.1. Biomaterials in Bone Tissue Engineering

Biomaterial is one of three key elements in bone tissue engineering, and it forms the skeleton for tissue regeneration. Biomaterials in bone regeneration can be sub-characterized into inorganic materials, organic materials, and composite materials.

#### 3.1.1. Inorganic Materials

Inorganic materials include medical metal materials and non-metal materials, which are characterized by high mechanical strength and are not easily deformed and degraded. Some require secondary surgery to remove.

Metal materials are ideal for bone repair in terms of load-bearing bone defects due to their remarkable mechanical properties. Metal-based biomaterials include titanium-based alloys, tantalum-based alloys, cobalt-based alloys, and magnesium-based alloys. Currently, titanium and its alloys are widely accepted in clinical application. The titanium-based alloys used in clinics are represented by pure Ti and titanium alloy Ti6Al4V [43]. Pure Ti has sufficient corrosion resistance in a physiological environment, but its poor strength and resistance limit its further clinical utilization. Compared with pure Ti, Ti6Al4V has optimal mechanical strength, flexibility, and fatigue resistance. Compared with various other metal materials, the elastic modulus of titanium-based alloy is highly relative to native bone, which is appropriate for applications in the field of orthopedics [44]. However, titanium lacks the ability to resist corrosion and bind to bone, so it is often required to add surface coatings to enhance its biological activity and corrosion resistance, including bio-adhesive coatings and composite coatings [21,45].

In 1940, Werman invested pure tantalum in the field of orthopedics as the pioneered biological material after titanium [46]. Pure tantalum was reported with no adverse reactions as a human implant. However, as its elastic modulus differs greatly compared to the host bone tissue, poor osseointegration is a result. Thereafter, porous tantalum was developed, and results showed outstanding capability to promote bone fusion and favorable orthopaedic capability [47,48,49,50]. Porous tantalum has an interconnected structure, and its particular porous architecture greatly contribute to its comparable elastic modulus to human cancellous bone and cortical bone. Porous tantalum is suitable for bone and joint replacement. Compared with titanium alloys, porous tantalum can promote cell adhesion and proliferation of bone marrow mesenchymal stem cells (BMSCs) and regulate the expression of osteogenic genes such as alkaline phosphatase (ALP), type I collagen, osteonectin and osteocalcin via activation of MAPK/ERK signaling pathway, and BMSCs osteogenic differentiation in vitro [51].

Both magnesium-based alloys and zinc-based alloys are biodegradable bone repair materials [52]. Magnesium is an essential element in the human body that is involved in cell metabolism [53]. Magnesium alloys have been extensively studied in repairing bone defects, attributing to their favorable degradability, plasticity, and mechanical strength, and it can avoid secondary surgery post implantation [54]. In a previous study, Mg^2+^ scaffold and hydroxyapatite scaffold were implanted into rabbit femur, respectively. Findings showed that both scaffolds had ideal biocompatibility [55]. The degree of degradation is rapid, and the ingrowth of new bone is obviously promoted, suggesting magnesium is a promising candidate for the treatment of bone defects.

In the group of medical metal materials, other types of materials are stainless steel, titanium, titanium alloys, and cobalt-based alloys. At present, many issue are presenting prior to clinical application. Difficulty in tissue integration was a major concern because the composition of metal materials varies from the composition of human tissue. In addition, most metal materials have an extremely low degradation rate, which requests additional operation for removal. Over the past 20 years, the modification of current medical metal materials with more stiffness, corrosion resistance, and biocompatibility was the goal in the research field of biodegradable metal and biological functionalization metal [56]. Using surface bioactive coating, drug coating and other related surface modification technologies was also found to further develop metal medical treatment appliance products.

Bioceramic has been studied in bone research based on their favorable biocompatibility, biodegradability, osteo-conductivity, and osteo-induction [57]. In bone tissue engineering research, bioceramics include hydroxyapatite (HA), β-tricalcium phosphate (β-TCP), biphasic calcium phosphate (BCP), bioglass, etc. Blank bioceramic powder cannot be directly used in repairing bone defects due to fast degradation and loss. Therefore, various porous three-dimensional tissue engineering scaffolds were prepared and proved to have sufficient mechanical support, nutrient exchange, and induction of tissue ingrown, suggesting bioceramics in treating large-size bone defects [58].

The doping of metal ions or bioactive ions opens a new avenue for the utilization of bioceramics. Human bone morphogenetic protein-2 (BMP-2)-coated bioceramic scaffolds promotes osteo-induction and bone remodeling. A calcium silicate/calcium phosphate scaffold was developed with macropores and micropores and loaded BMP-2 [59]. In this study, authors found that the microporous scaffold retained the secondary structure and biological activity of RhBMP-2, and the local release of BMP-2 promoted the formation of new bone. Another study used biphasic calcium phosphate combined with BMP2-precipitated layer-by-layer assembled biomimetic calcium phosphate particles (bone morphogenetic protein-2 coprecipitated biomimetic calcium phosphate particles, BMP2-cop-BioCaP) for repairing rat calvarial defects [60]; BMP2-cop-BioCaP in improving bone formation was comparable to the most often used osteoinducer in clinical practice—autologous bone.

#### 3.1.2. Natural Biomaterials

Natural biomaterials, including collagen, chitosan, sodium alginate silk fibroin, and hyaluronic acid can simulate natural bone extracellular matrix, followed by biodegradation into carbon dioxide and water in vivo. Natural biomaterial is widely used in the preparation of bone tissue engineering scaffolds based on its convenient material acquisition, good plasticity, and good biocompatibility.

Collagen is the major component in skin, bone, tendon, and ligament, and has high swelling rate and low antigenicity, which is ideal natural material in bone tissue engineering. However, its poor mechanical properties limit its direct use as a substitute for bone, therefore, composite scaffolds of collagen together with high physical strength has received attention [61]. Chitosan is a natural cationic carbohydrate material that is partially deacetylated from chitin. It is a non-antigenic, non-toxic, biodegradable material with certain biological functions. However, as chitosan is insoluble in water, has fast biodegradation in vivo, and has poor compatibility with blood, its potential for bone regeneration is limited. Researchers realized the function of chitosan structure by compounding chitosan with various other materials, such as HA [62]. Such a combination solves the limitation of its application in bone defect repair. Fibrin is the major component of the extracellular matrix, and it has been proved to mediate intercellular signal transduction and interaction [63]. A 3D fibrin/sodium alginate scaffold was successfully constructed on a titanium plate and the finding showed that such modification is capable of improving cell adhesion, proliferation, and subsequent differentiation of human mesenchymal stem cells into osteoblasts [64]. At present, most of the studies on the preparation of fibrin in bone tissue engineering scaffolds have failed in pre-clinical test via small and large animal models.

Deproteinized bovine mineral matrix (Bio-Oss) is naturally deproteinized from the mineral fraction of bovine cancellous and cortical bone, which retains fine trabecular structure and internal pores, providing favorable conditions for osteoblast ingrowth and angiogenesis [65,66]. Bio-Oss bone contains more carbonate to facilitate autologous osseointegration to achieve the required mechanical strength and stiffness [67]. Clinically, Bio-Oss bone powder is mixed with normal saline or the patient’s own blood to form a paste on the bone defect, and it can be precisely sized with easy operational purpose. However, few studies demonstrated that Bio-Oss materials are difficult to be absorbed over time [68], while it needs to be mixed with normal saline or venous blood for use. Over bleeding with the flow dispersion of Bio-Oss can increase surgical difficulties, resulting in severe loss of bone meal over the bone grafting process and osteogenesis [69].

Autologous tooth graft material (AutoBT) and autologous dentin particles can be prepared via extracting unretained teeth from host patient, followed by implantation into the patient’s bone defect as a graft. Due to its favorable biocompatibility, osteoconductivity and osteoinductive effects, and no immune rejection, it is expected to have a promising clinical outcome compared to many other commercial products [70]. However, the mechanism of how dentin induced osteogenesis is unclear, while preparation processing is cumbersome and time-consuming, which may limit its wide application in clinical practice [71].

#### 3.1.3. Synthetic Polymer

Synthetic polymer materials are widely studied for bone regeneration, including commonly used polylactic acid (PLA), polyglycolic acid (PGA) and polylactic acid-glycolic acid copolymer (PLGA). Polymethyl methacrylate (PMMA) bone cement was the bone cement utilized in clinical practice, due to its fast-setting speed and better mechanical strength. However, it is known to cause mild damage to bone surrounding tissue, and its monomer has proven biological toxicity [72,73,74,75]. Additionally, the low biodegradation rate of PMMA in the defect area can negatively affect the growth of new bone, resulting in being non-conducive to the regeneration and repair of bone defect in the future clinical use [73].

In recent years, due to the rapid development of polymer materials, polyetherketone ketone (PEEK), as a new biocompatible high-performance polymer, has been approved by FDA as an implant device and gradually applied in the biomedical field. Within the elastic modulus of 4.5 GPa, PEEK is closer to that of human bone, which can meet the normal physiological needs of the human body [76]. PEEK is an organic thermoplastic polymer with good biocompatibility, heat resistance, corrosion resistance, etc. leading to “the most promising material in the 21st century”. Few organic synthetic polymer materials, such as: polymethyl methacrylate, polyurethane, polylactide, polyglycolide, polycaprolactone, etc., are remaining as the research hotspots of bone tissue engineering scaffolds, but these materials are not widely used in biological applications. Degradability, biocompatibility, and other aspects cannot meet the requirements of an ideal scaffold, so the research on the modification of materials is particularly important [77].

Over 20 new bone graft substitutes have been used to treat different types of bone defects in the last decade (Table 1). Most bone graft substitutes are hydroxyapatite and deproteinized mineral matrix materials.

### 3.2. Cells and Stem Cells in Bone Repair

In the field of tissue engineering and regenerative medicine, stem cells that can be isolated from tissues such as bone marrow or adipose tissue [103] have been used for the treatment of bone defects for years.

Bone marrow mesenchymal stem cells (BMSCs) are a heterogeneous population of cells obtained from the bone marrow stromal fraction [104]. They have high self-proliferation and multi-directional differentiation potential, which are recognized as favorable cell types in bone tissue engineering [105,106,107]. In in-vitro studies, BMSCs can rapidly amplify and differentiate into various mesodermal lineages, such as adipocyte, chondrocytes, and osteocytes, greatly contributing to the regeneration of osteochondral tendon, fat, and muscle [108,109]. In 2001, three patients having large bone defects were successfully treated as BMSCs for the first clinical trial [110]. However, a relatively low abundance of BMSCs in vitro expansion reduced post-translational survival and immunomodulatory BMSCs, which are regulatory and logical challenges [111]. In addition, the age of donors is a key factor for cell survival and differentiation that should be considered in both basic and clinical evaluations [112].

Adipose-derived stem cells (ADSCs) are a population of stromal cells that can be isolated from adipose tissue, with comparable morphology and phenotype to BMSCs. ADSCs extraction is easy, and cell proliferation is not affected by patient’s age with multi-functional differentiation. ADSCs are capable to differentiate into osteoblasts, chondrocytes, and adipocytes [113,114,115]. Post being implanted into the body, adipose-derived mesenchymal stem cells can adapt to the physiological, pathological, stress, and other microenvironments of the local area and maintain osteogenic activity [116]. In a study, a 7-year-old pediatric patient having post-traumatic calvarial defects was successfully treated with autologous ADSCs, fibrin glue, and biodegradable scaffold. Postoperative new bone formation as well as relatively complete calvarial continuity was formed based on computed tomography analysis [117,118].

With the continuing development of bone tissue engineering, researchers are seeking for other potential seed cells to repair bone defects. At present, many seed cells with osteogenic activity are still in experimental stage, such as embryonic stem cells (ESCs) [119], periosteum-derived cells (PDCs) [120], dental pulp stem cells (DPSCs) [121], human amniotic mesenchymal stem cells (HAMSCs), etc. Moreover, aiming to achieve the best repair outcomes, researchers are also working on the combination of varying stem cells. HAMSCs and HBMSCs were used together, of which having osteogenic ability but different advantages [122]. Results of co-culture showed that the mineralized nodules formed in the co-culture system were more significant compared to that in single culture, while all osteogenic markers were also significantly up regulated [122].

### 3.3. Active Factors of Bone Tissue Engineering

Growth factors play an auxiliary role in bone tissue engineering. Nowadays, viral or non-viral vectors are widely used to deliver growth factors and promote osteogenesis with accelerated vascularization in the defect areas [123]. Growth factors used for bone defect repair include bone morphogenetic protein-2 (BMP-2), fibroblast growth factor-2 (FGF-2), and vascular endothelial growth factor (vascular endothelial growth factor, VEGF) etc. BMP-2 is a member from the transforming growth factor-β protein family, responding for osteogenesis in vivo. BMP-2 simultaneously promotes bone regeneration and stimulates angiogenesis in the defect area, while it is approved by FDA as osteoinductive growth factor in clinical practice. FGF-2, also known as basic fibroblast growth factor, is a canonical FGF that belongs to the FGF-1 subfamily [124]. FGF is involved in osteoblasts proliferation and differentiation [125], angiogenesis, and in signal transduction within the cell membrane of bone progenitor [126]. However, concentrations of FGF can alter its effects: A high dose of FGF was found to inhibit bone formation while a low dose increase bone formation [127,128]. VEGF promotes angiogenesis and can also regulate the osteogenic process. Osteoblast-derived VEGF can stimulate cell differentiation of mesenchymal stem cells into osteoblasts and inhibit its differentiation into adipocytes, as a key role maintaining bone homeostasis [129].

In bone tissue engineering, researchers attempt to combine two or multiple materials, aiming for multi-functions and overcoming disadvantages of a single material. Scientists use bionics to improve the biological properties via loading bioactive factors, as well as incorporating varying materials to develop composites with more gradient and a controllable degradation rate. For example, a biocompatible and resorbable scaffold is produced with minimal tissue rejection and better bone tissue growth [130]. Naudot et al. [131] prepared honeycomb polycaprolactone (PCL)-nano-hydroxyapatite (nHA) composite scaffold via 3D electrospinning and printing technology. The combination of BMSCs with such scaffold was found to significantly improve bone regeneration and bone mineralization in a rat calvarial defect model [131]. Moreover, mixed polymethyl methacrylate together with nano-scale tricalcium phosphate and hydroxyethyl methacrylate in different proportions was investigated to produce novel porous tricalcium phosphate/hydroxyethyl methacrylate/polymethyl methacrylate [132]. It is pre-clinically proved to be a new bone substitute prior to clinical tests.

Silk fibroin/chitosan/nanohydroxyapatite (SF/CS/nHA) 3D scaffolds combined with bone marrow stromal cells were used to repair the rabbit radius defect model [133]. In comparison with blank SF/CS/nHA scaffolds, BMSCs-loaded SF/CS/nHA scaffolds had a better stiffness on the repair of radial bone defect [133]. Adipose-derived mesenchymal stem cells on 3D-printed titanium scaffolds showed significantly faster cell proliferation and osteogenicity [134]. BMP-2 combined with a scaffold is an effective method to promote osteoinduction and bone remodeling. Zhang et al. [59] developed a calcium silicate/calcium phosphate scaffold with macropores and micropores, loaded with human bone morphogenetic protein-2, which could maintain its structural integrity and biological activity, as well as controlled release. In addition, local delivery of BMP-2 loaded with microporous calcium phosphate can further accelerate osteoclast resorption and promote new bone formation [135]. BMP-2-loaded PLGA nanoparticles coating on the surface of HA scaffolds were found to be uniformly distributed on the surface of the scaffold, with sustained release of BMP-2 over 30 days. Additionally, PCL-BMP-2/PLGA nanoparticles can improve cell proliferation, adhesion, and osteogenic differentiation both in vitro and in vivo [136].

### 3.4. Preparation of Bone Tissue Scaffolds

In bone tissue engineering, novel approaches in terms of scaffold preparation methods are widely investigated. Hydroxyapatite/chitosan (SrHAP/CS) nanocomposite scaffolds with different concentrations of strontium were reported to be prepared by freeze-drying method [137]. Results showed that the different composites had good cytocompatibility and promoted cell adhesion, integration, and proliferation of hBMSCs. Delivery of Sr^2+^ can significantly enhance cell proliferation and osteogenic differentiation. In addition, due to the synergistic effect of Ca^2+^ and Sr^2+^, a high-concentration Sr-loaded scaffold was found to be optimal in osteogenic induction. Nano-hydroxyapatite/chitosan scaffolds loaded with slow-release microspheres of Mutong saponin D were also reported to be prepared by freeze-drying. Findings also demonstrated that scaffolds improve cell adhesion, proliferation, and osteogenic differentiation of osteoblasts [138]. Recently, a co-culture of rabbit adipose-derived mesenchymal stem cells (ADSCs) together with double-cell sheet DCS complexes was carried out. DCS-PLL-CHA, a coral hydroxyapatite (CHA) composite scaffold modified with DCS and polylysine (PLL), was prepared by soaking and vacuum freeze drying. Results show that PLL can effectively promote the proliferation and differentiation of ADSCs, and DCS-PLL-CHA vascularized tissue engineered bone has the potential to promote bone regeneration and bone remodeling, which can be used to repair large bone defects [139].

Thermally induced phase separation is a method to prepare polymeric nanofibrous materials that resemble natural extracellular matrices. The first step is to mix a polymer with a liquid or solid diluent of high boiling point and low molecular mass to form a homogeneous solution at high temperature, followed by casting a mixture solution into the desired shape, while lowering the temperature enables solution phase-separation. These solvents were extracted to remove the diluent and finally freeze-dried to obtain the pore structure. Macroporous nanofiber scaffolds-polylactic acid-glycolic acid copolymer microspheres/nanofibers (BMP-2@MS/NF) loaded with human bone morphogenetic protein 2 using cloud point thermally induced phase-separation method was recently reported [140]. Studies have shown that the composite scaffold can enhance the adhesion and proliferation of mouse primary osteoblasts and promote the repair of rat calvarial defects. PLLA/1,4-dioxane/H_2_O ternary system was used to prepare macroporous PLLA scaffolds, via treating with acetone and immersed in chitosan solution for modification [141].

Electrospinning technology is a direct and continuous method for preparing polymer nanofibers [142]. Due to its simple production process, scaffold materials with nanoscale fibers can be synthesized, and the morphology of the fiber surface can be adjusted by changing the conditions of electrospinning. Electrospun PCL scaffolds have been widely investigate based on their great potential in mimicking the structure of a native extracellular matrix (ECM). However, relatively small pore size and low bioactivity of the scaffolds limit tissue regeneration. PCL (polycaprolactone), PCL/PEG (polyethylene glycol), and PCL/PEG/ATP (nano-attapulgite) scaffolds were produced by electrospinning, and water-soluble PEG fibers were removed by washing to increase scaffold pores rate [143]. This study was the first to show that scaffolds after PEG removal had better cell infiltration compared to non-washed scaffolds. Compared with PCL scaffolds, ATP-doped electrospun PCL scaffolds further improved bone regeneration in rat calvarial defects [143]. Enhanced osteogenesis and bone repair were found to be associated with PCL/ATP-activated BMP/Smad signaling pathway [143]. Electrospinning also provides an opportunity to prepare nanofibrous scaffolds that mimic the structure of a natural extracellular matrix (ECM) with high porosity and large specific surface area. However, the small pore size in the range of 10~50 µm in traditional electrospinning-prepared scaffolds normally limits cell infiltration and tissue regeneration. To achieve satisfactory results in tissue engineering, it is necessary to combine conventional, coaxial electrospinning and advanced techniques to produce better 3D structures with larger pores and open space. Fabricating 3D structures for bone tissue regeneration remains challenging.

Within the rapid development of 3D-printing technology in the last decade, this technology has been widely used in tissue engineering. 3D printing, also known as “additive manufacturing”, is an advanced manufacturing process. Based on the 3D model data of computer-aided design, it can quickly manufacture entities that are highly consistent with the design model [144].

Controlled internal macroscopic structure and the design of the shape matching are always considered as challenges. Scientists use the characteristics of 3D printing to produce different organs and tissues, which can largely solve the problem of insufficient organ donors [145]. At present, the application of 3D printing technology in orthopedics mainly focuses on the preparation of orthopedic scaffolds and macroscopic skeleton with a modified internal porous structure [146]. Advanced methods such as indirect 3D printing and 4D printing are also considered to be utilized in bone tissue engineering. Additionally, 3D-printing technologies can be divided into laser or high-energy-density heat source, including photo-curing (Stereo lithography appearance, SLA), and selected laser sintering (Selected laser sintering, SLS). Jet-based forming technology such as fused deposition modeling (Fused deposition modeling, FDM), 3D printing (3D printing, 3DP), and direct ink 3D printing (DIW) is in another group of 3D printing (Table 2) [144]. However, now existing 3D printing technology is known to enable cell fusion during the printing process, other than 3D bioprinting to produce cell-laden hydrogel structures. Therefore, advanced 3D-printing techniques should be invented to enable simultaneous scaffold fabrication and cell fusion.

## 4. Adjuvant Therapy

### 4.1. Physiotherapy

Physical intervention and promotion of the bone repair through exogenous stimuli such as light, heat, electricity, and magnetic fields are also widely investigated in basic science and clinical therapy (Figure 2).

Photothermal therapy (PTT) has been used to promote tissue regeneration as it has low destructiveness, superior tissue penetration, non-invasiveness, and controllability. Mild heat (40~43 °C) is proved to effectively promote bone regeneration. A study synthesized porous AuPd alloy nanoparticles as a hyperthermia agent and conducted in-situ bone regeneration of critical calvarial defect in rats by photothermal therapy. Results found that after being swallowed by cells, almost 97% of the cranial defect area (8 mm in diameter) was covered by the newly formed bone after 6 weeks of PTT [156]. Exogenous electrical stimulation produces progressive cell attachment, proliferation, and differentiation through cell–cell and cell–scaffold interactions [157,158,159]. Therefore, the introduction of appropriate electrical stimulation may have a positive significance for bone repair and regeneration. Maharjan et al. [160] found that the electrical stimulation on PCL/polypyrrole (PPy) scaffolds enhanced cell adhesion, growth, and proliferation of MC3T3-E1, while calcium and phosphorus deposition on the scaffold surface was also found significantly increased. This finding provides a new strategy for preparing conductive scaffolds with higher bioactivity and osteogenic differentiation ability under electrical stimulation. Another important role in the regulation of cellular responses is magnetic field. A study found that the polycaprolactone/magnetic nanoparticle magnetic nanocomposite scaffold, which was assisted by an external static magnetic field (SMF), can synergistically act on primary mouse calvarial osteoblasts and stimulate angiogenesis [161]. Magnetic field stimulation was also found to accelerate tissue binding of the scaffold to host bone with increased calcium deposition and bone density, thereby accelerating the healing of critical defects in the mouse calvaria.

### 4.2. Other Techniques Involved in Bone Tissue Engineering

#### 4.2.1. Exosome

Exosomes (Exos) were first discovered in the late 1980s, when sheep reticulocytes secreted membrane proteins through exocytosis during maturation, which were named “Exosomes” as the existence of extracellular space [162]. Exosomes, with a diameter of 30 ~150 nm, are now known to be secreted by a variety of cells (such as mesenchymal stem cells, dendritic cells, epithelial cells, adipocytes and B cells) and widely exist in blood, urine, saliva, and other body fluids [163]. Exosomes are mainly composed of phospholipid bilayers containing (mRNA, miRNA, DNA, lipids, and proteins) [164]. Proteomics showed that exosomes contained tetraspnin proteins (CD63, CD81, CD9) and antigen presenting proteins (MHCI and MHCII) involved in immune response [165]. Exosomes can transport their contents to target cells, thus playing a role in intercellular communication, influencing microenvironment, and regulating physiological functions of cells. Current studies have shown that exosomes participate in tissue repair and regeneration, disease diagnosis, tumor invasion and metastasis, immune regulation, and drug-targeted transport (Table 3) [166].

#### 4.2.2. Microneedling

Microneedle (MN) is micron-scale needle that can penetrate the skin cuticle, form microporous channels, and promote drug penetration. Recently, MN has attracted great attention in application thanks to its low pain, high safety, and outstanding therapeutic effects [184]. MNs are used to treat many bone disorders [185,186,187]. Studies have shown that microneedles can be used for the delivery of alendronate and improve the bioavailability of the drug. Katsumi et al. designed a self-dissolving ALN microneedle patch based on hydroxyapatite. Alendronate was loaded in the whole microneedle made of hyaluronic acid. Through the application of the microneedle array in the rat model of osteoporosis, findings showed that transdermal administration of alendronate can modulate bone resorption of osteoclasts in treating osteoporosis and achieve approximately 90% bioavailability [188]. In order to avoid skin irritation and reduce the loss of drug residues at the base of MNs, alendronate microneedle was further modified by immersing alendronate onto the tip of the needle only [189]. Human parathyroid hormone (1-34) (PTH) is a polypeptide that can be used to treat osteoporosis and promote bone healing [190]. However. frequent injection is not always accepted by patients. Therefore, alternative administration methods using PTH (1-34)-coated microneedle patch was investigated in the treatment of osteoporosis [191] with phase II trial completed [192].

Clinical pharmacokinetics and pharmacodynamics of PTH (1-34)-coated microneedle patch were examined [186]. Scientists found that ZP-PTH patch had sustained, rapid, and efficient delivery of PTH with short plasma exposure time and significantly increased bone mineral density in the lumbar spine and hip. Interestingly, ZP-PTH also increased hip bone mineral density over 6 months, and this effect was not observed with subcutaneous injections [186]. Some studies have used hyaluronic acid as a microneedle shell to prepare soluble microneedle for efficient percutaneous delivery of PTH [193]. Microneedle has sufficient mechanical strength and penetrates the stratum corneum, epidermis, and upper dermis. At the same time, drugs in the microneedle have high stability. PTH (1-34) with pharmacological activity can be effectively transmitted to bone through transdermal absorption. The application of MN loaded with parathyroid hormone enhances bone formation [193]. Abaloparatide (TymlosTM) is a synthetic peptide analogue of human parathyroid hormone-related protein that can increase bone formation and reduce the risk of fracture in postmenopausal women with osteoporosis [194]. At present, Radius is still developing a transdermal preparation of abaloparatide administered by microneedle patch [195]. In addition, a study showed that the cyclic peptide drug salmon calcitonin (SCT)-coated MN transdermal patch can promote osteoblast proliferation and differentiation, thus replacing traditional subcutaneous and nasal administration. The surface of the microneedle is coated with SCT dry powder drug formulations, which can be dissolved when inserted into the skin, improving the bioavailability of the drug [196]. Based on this design, studies have used two dissolving microneedle arrays (DMNAs) to deliver salmon calcitonin, avoiding sharp biological hazards. Compared with the traditional transdermal gel patch, DMNA can significantly improve the therapeutic effect of sCT [197], providing a feasible scheme for the treatment of bone diseases in the future. In addition, a method of combining conductive MN and ITP to achieve local anesthesia of teeth was proposed. The study found that under the rabbit-inhibitor model, the drug can quickly pass through the oral mucosa and alveolar bone to reach the tooth sensory supply nerve to produce an anesthetic effect [198]. Compared with the traditional needle, the micro needle overcomes people’s fear of the needle. In conclusion, MN has great potential in the treatment of bone diseases.

## 5. Conclusions and Future Perspectives

In conclusion, bone defect repair is still a challenge faced by orthopedics. The etiology of bone defects mainly includes two aspects, one is congenital factors, the other is acquired factors, and how to prevent bone defects is mainly caused by acquired factors. Acquired factors can be skeletal injuries caused by external forces or the sequelae of various diseases and operations. The prevention and management of these injuries is mainly to reduce the risk of bone stress injury. The management of bone stress injuries depends on their location and the risk of healing complications. Bone stress injuries in low-risk sites can usually be adjusted for healing with physical exercise, followed by a gradual reintroduction of the activity [199]. Treatment options for bone stress injury at high-risk sites include non-weight-bearing immobilization, medical therapy, or surgery. Although the underlying mechanism of bone stress injury is currently unclear, the prevailing theory is that maintaining a balance of bone metabolism is beneficial to reduce the accumulation of bone damage [200]. These skeletal injuries result in damage to the plasma membrane of skeletal muscle cells, while mitochondrial function plays a crucial role in the self-repair of such plasma membrane [201]. Few studies reported that aging of mitochondria in skeletal muscle can be prevented by limiting the caloric intake in animal models; it can also accelerate the internal ability of skeletal muscle to repair by supplementing some endogenous proteins, such as BMP-2 [202,203].

The emergence of bone tissue engineering technology made remarkable progress in the study of new materials and significantly promoted the progress of bone defect treatment. However, low bionic materials, poor biocompatibility, cell migration, adhesion, and proliferation are still remaining as the issues. The precise control of cell differentiation, genes expression and growth factors, as well as safety of clinical applications are still needed to be further investigated. With the cross-integration and development of multidisciplinary concepts and technologies in the field of bone repair, such as cell biology, immunology, materials science, and manufacturing, bioactive bone-like tissues can be constructed by constructing new bone repair substitutes or directly in vitro. Regenerative repair of bone defects is believed to apply in clinical practice in the near future.

## Figures and Tables

**Figure 1 pharmaceuticals-15-00879-f001:**
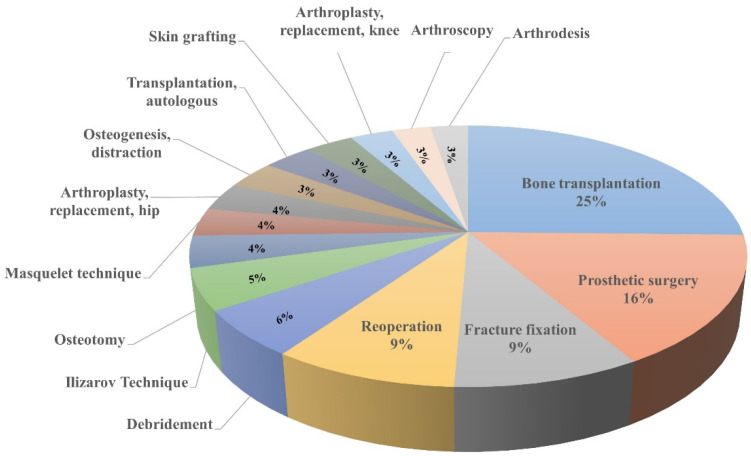
Surgical treatment for repairing bone defects. Bone transplantation, prosthetic surgery, reoperation, and fracture fixation are listed as the top four surgical treatments.

**Figure 2 pharmaceuticals-15-00879-f002:**
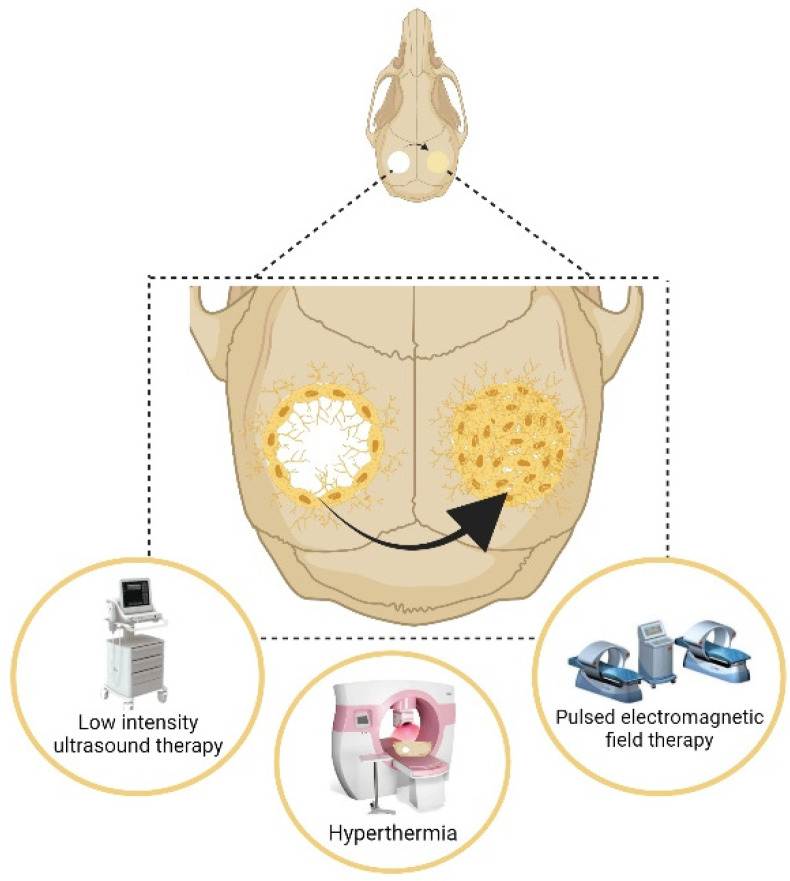
Different exogenous stimulation promotes the recovery of bone defects.

**Table 1 pharmaceuticals-15-00879-t001:** Bone substitutes used in bone defects.

Bone Substitute	Company and Location	Composition	Indication	Pore or Particle Size	Incorporated	References
k-IBS^®^	AritMedical, Spain	Hydroxyapatite (HA) and β-Tricalcium Phosphate (β-TCP) The HA/-TCP ratio was 3/1	Solitary enchondroma in the hand bones			[78]
InterOss^®^	Sigma, USA	Mixing bovine hydroxyapatite granules to porcine derived collagen in water in 9:1 ratio (by weight)	Fill or reconstruct periodontal and bony defects in the mouth			[79]
Bontree^®^	HudensBio Co., Gwangju, Korea	OCP and HA mixed at a weight ratio of 80:20	Alveolar ridge or sinus augmentation	0.5–1 mm		[80]
CustomBone^®^	DePuy Synthes, USA	60% calcium sulfate and 40% HA	Human tibial fractures			[81]
Traumacem™ V+	DePuy Synthes, USA	Acrylic bone cement in conjunction with ceramics consisting of 45% PMMA, 40% zirconium dioxide, 14.5% hydroxyapatite, and 0.5% benzoyl peroxide	Calcaneal fracture			[82]
Vitoss BA^®^	Stryker, Kalamazoo, USA	β-TCP particles bonded on a collagen matrix supplemented with 20 wt% 45S5 bioactive glass particles	Metaphyseal bone defect	90–150 μm		[83]
HydroSet™		Tetracalcium phosphate (73%), dicalcium phosphate anhydrous (27%) and Na_2_HPO_4_, NaH_2_PO_4_ and Polyvinylpyrrolidone	Bone defect, skeletal fractures, hip replacements			[84]
MIIG^®^ X3	Wright Medical Technolog, Inc., Arlington, TN	Calcium sulfate	Comminuted calcaneal fractures			[85]
Calciresorb C35^®^	Ceraver, USA	Macroporous biphasic calcium phosphate ceramic granules (HA/TCP = 65/35)	Femoral bone defect	6 mm	Mesenchymal stem cells	[86]
ChronOS^®^	Depuy Synthes, Massachusetts, USA	TCP	Bone defect	5.03 ± 1.90 μm		[87]
Graftys^®^	Aix-en-Provence, France	α-tricalcium phosphate, dicalcium phosphate dihydrate, monocalcium monohydrate, calcium-deficient hydroxyapatite, hydroxypropyl methyl cellulose	Bone defect			[88]
Cerament^®^		60% calcium sulfate (CS) and 40% HA	Acute traumatic depression fractures of the proximal tibia			[89]
Bio-Oss^®^	Geistlich, Wolhusen, Switzerland	90% DBBM extracted from cattle and 10% highly purified porcine collagen matrix	Alveolar bone resorption	0.25–1 mm		[90]
Healos^®^	DePuy Orthopaedics, Inc.	Osteoconductive sponge made of collagen fibers coated with hydroxyapatite	Bone defect		Recombinant human bone morphogenetic protein-2	[91]
SINTlife^®^	Fin-Ceramica, SpA, Faenza, Italy	Nano-structured Mg-enriched hydroxyapatite	Bone defect	30–40 nm		[92]
DBSint^®^	Fin-Ceramica, SpA, Faenza, Italy	Nano-structured Mg-enriched hydroxyapatite and human demineralized bone matrix	Bone defect			[92]
OsteoSet^®^2 demineralised bone matrix	Wright Medical Group Inc., Memphis, Tennessee, USA	DBM particles homogenously dispersed throughout surgical-grade calcium sulphate	Large osteonecrotic lesions of the femoral head	3.5–4.8 mm		[93]
OCS-B^®^		Calf bone powder, bone inorganic material in calf bone	Bone defect	0.2–1 mm		[94]
BoneSource^®^	Stryker Orthopaedics, Mahwah, New Jersey	An equimolar mixture of tetracalcium phosphate and anhydrous dicalcium phosphate	Bone defect	33.4 ± 6.2 μm		[95,96]
Ostim^®^	aap biomaterials GmbH, Dieburg, Germany	Nanosized HA and calcium sulphate	Metaphyseal osseous volume defects	19 nm		[97,98]
Cortoss™	Orthovita^®^, Malvern, USA	Acrylic resin reinforced with glass ceramic particles, 30% copolymerizing organic components and 70% glass-ceramic fillers	Calvarial defects	148.4 ± 70.6 μm		[96,99]
Calcibon^®^	Biomet-Merck Biomaterials GmbH, Darmstadt, Germany	61% alpha-TCP, 26% calcium-hydrogeno-phosphate, 10% calcium-carbonate and 3% hydroxyl-apatite	Acute traumatic compression vertebral fracture without neurological deficit	41.6 ± 22.0 μm		[96,100]
α-BSM^®^		Apatitic calcium phosphate	Articular depression fractures	12–14 nm		[101]
Norian SRS^®^		Monocalcium phosphate, tricalcium phosphate, calcium carbonate and sodium phosphate	Distal radial fracture	47.2 ± 21.9 μm		[96,102]

**Table 2 pharmaceuticals-15-00879-t002:** Principles and applications of 3D printing of bone tissue engineering scaffolds.

Principle	Method	Advantage	Disadvantage	Materials and Bio-ink	Application	Reference
Laser or high energy density heat source	Stereo lithography appearance, SLA	Fast processing speed; high maturity; high precision	High cost; software operation difficulty; high environmental requirements	Hydroxyapatite; calcium chloride and diammonium hydrogen phosphate	parietal bone; cancellous bone repair;	[147,148]
	Selected laser sintering, SLS	Wide selection of materials; without add organic adhesives;	High cost and low efficiency;	titanium alloy; alendronate-collagen; PVA-HA	segmental bone defects; alveolar bone implant therapy;	[132,149,150]
Spray forming technology	Fused deposition modeling, FDM	Low cost; simple manufacturing; wide application range;	Low precision; rough surface; slow speed	PLGA; PCL-deferoxamine	cancellous bone formation; segmental bone defect	[151,152]
	3D printing, 3DP	Printable active substance; prepared complex scaffolds;	Drying time is long; ink is easy to deteriorate	HA powders, air jet milling powders, spherical powder	Mandibular defect;	[153]
	Direct ink writing 3D printing (DIW)	fast printing speed; easy operation; low cost; high precision;	Low molding accuracy; easy to deform [154].	PLA/CA	Craniomaxillofacial Reconstruction	[155]

**Table 3 pharmaceuticals-15-00879-t003:** Applications and targets of exosomes from different sources in the treatment of bone defects.

Origin of Exosomes	Target	Application	References
human mesenchymal stem cells exosomes	MiR-29a	mice with nonhealing skull defects	[167]
Osteogenic Human exosomes	MiR-199b/MiR-218/MiR-148a/MiR-135b/MiR-221	human bone marrow-derived mesenchymal stem cells; osteoblast cells	[168]
Human bone marrow stromal/stem cell exosomes	MiR-196a/MiR-27a/MiR-206	bone formation in Sprague Dawley (SD) rats with calvarial defects; osteoblast cells	[169]
human-induced pluripotent stem cell-derived mesenchymal stem cells exosomes	Akt/p-Akt	human bone marrow-derived mesenchymal stem cells	[170]
stem cells from apical papilla-derived exosomes	MiRNA-126-5p/MiRNA-150-5p	the mandibular defects of diabetic rats	[171]
mesenchymal stem cells exosomes	green fluorescent protein (GFP)	old male C57BL/6 mice	[172]
Bone marrow mesenchymal stem cells exosomes	Smad/RUNX2	acute rotator cuff rupture in rabbits	[173]
M2 macrophagy-derived exosomal	MiRNA-26a-5p	Osteogenic differentiation of BMSCs	[174]
Exosomes of human umbilical vein endothelial cells	Pd-1 on the surface of T cells	callus formation and fracture healing in a murine model	[175]
Exosomes of M2 macrophages	MiR-690 / IRS-1/TAZ	bone marrow mesenchymal stem cells	[176]
Exosomes of bone mesenchymal stem cells	MiR-1260a	calvarial defect rat model.	[177]
Exosomes derived from mesenchymal stem cells	MiR-21/NOTCH1/DLL4	skull defects in rats.	[178]
Exosomes derived from mesenchymal stem cells	Acvr2b/Acvr1	rat skull defect model	[179]
Exosomes derived from bone marrow mesenchymal stem cells	RAB27B/SMPD3	Human bone marrow mesenchymal stem cells; osteogenic cells; SD rats	[180]
Exosomes derived from bone marrow stem cells	NF-κB	BMSC. rat balloon models and rat femoral borehole models	[181]
Exosomes of mature dendritic cells	large tongue suppressor kinase 1 (LATS1)	femoral bone defect in athymic rats	[182]
Exosomes derived from bone marrow stromal cells	MiR-146a	human umbilical vein endothelial cells; distal femur defect in rats.	[183]

## Data Availability

Not applicable.

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
