# Peer review of "Bone Tissue Engineering in the Treatment of Bone Defects"

_pharmaceuticals, 2022, doi:10.3390/ph15070879_

Round 1

Reviewer 1 Report

The topic of this review article is of interest for the researchers in the field of bone tissue repair, hard tissue engineering and regeneration. The aim of this study is to summarize and discuss new techniques and advances in develop artificial bone. I particularly like the images I recommend the authors to answer few questions below to enable this manuscript more suitable for publication in pharmaceutics. 

Q1: Figure legends of Figure 2 should be added more information. 

Q2: Please add some necessary references,such as on page 1,line 27-33. 

Q3: 3D printing materials and bio-ink can be considered to add and expend the text. 

Q4: Few spelling errors and typos which needs to be revised. “limintation” should to be “limitation”, “Sr2+” should to be “Sr2+”, the authors need to revise the whole manuscript carefully.

Q5: The “conclusions and future perspectives” need to have more comprehensive discussion.

Author Response

Q1: Figure legends of Figure 2 should be added more information.
A1:Thank you for reviewer’s comments. Based on editor’s suggestion, we deleted Figure 1 in the 
manuscript and revised Figure 2 as the Figure 1 in the manuscript. Detailed figure legend for Figure 1 
is added on page 3, line 92. 
Q2: Please add some necessary references,such as on page 1,line 27-33.
A2: We have added new references to the text on 1 page, line 42. “Reference [2]”
Q3: 3D printing materials and bio-ink can be considered to add and expend the text.
A3: We acknowledged for the suggestions. 3D printed materials and bio-ink are added into Table 2.
Q4: Few spelling errors and typos which needs to be revised. “limintation” should to be “limitation”, 
“Sr2+” should to be “Sr2+”, the authors need to revise the whole manuscript carefully.
A4: We apologized for the spelling and typos. We have double checked and corrected the full text.
“ limitation” page 3,line 97, “proliferation” page 4,line 145, “Moreover” page 13,line 392, “ Mg2+”
page 6,line 217, “Sr2+
, Ca2+”page 13,line 420-421.
Q5: The “conclusions and future perspectives” need to have more comprehensive discussion.
A5:We acknowledge these comments. We have revised the content in the “Conclusion and Future 
Perspectives.”on page 21, line 623

Reviewer 2 Report

·         Abstract is too short, reformulate and enrich

·         Also, add keyword tissue engineering

·         Add a paragraph in the introduction about how to care for bones, methods of prevention, and treatment before the need for organ transplantation

·         Line 52 add examples for these materials

·         Table 1 separate the citation in the new column as table 2 and 3

·         Line 124 adjust the citation

·         Add reference in lines 212, 231, 239, 268

·         Clear the role of nanocomposites or nanofiber reinforcements in polymer matrices in bone regeneration

·         Check the citation style all over the document

·         Sec 4.1 adjust the line spacing

·         Table 3, it should be MiR-, check the table

·         Add a paragraph about How to prevent bone damage

·         The review needs to be updated, most of the references are outdated, update to 2021, 2022

·         Check the outputs of all references, some lack page number or volume

·         Journal name abbreviated

Author Response

Reviewer 2

Comments and Suggestions for Authors.

  • Abstract is too short, reformulate and enrich

A1: Thanks for reviewer 2’s comments. The abstract has been corrected and extend due to the comments on page 1 line 15.

  • Also, add keyword tissue engineering

A2: Thank you for the comment. “tissue engineering” is added to keywords on page 1 line 30.

  • Add a paragraph in the introduction about how to care for bones, methods of prevention, and treatment before the need for organ transplantation

A3: We have added the introduction of daily care and prevention methods of bones in the Introduction on page 1and page 2, line 44-54.

  • Line 52 add examples for these materials

A3: We acknowledge these comments. We have paraphrased the passage on page 2, lines 70-74.

Table 1 separate the citation in the new column as table 2 and 3

A4: Thank you for your suggestion. We have adjusted Table 1.

  • Line 124 adjust the citation

A5: Thanks for your suggestion. We have double checked and revised the citation on page 4, lines 150-152, also in the reference list “[25, 26]”

  • Add reference in lines 212, 231, 239, 268

A6: Thank you for the comment. we have checked the entire manuscript and added required references.

lines 239,Page 6. “Literature [51]”; lines 256,Page 6. “Literature [56]”; lines 266,Page 7. “Literature [58]”; lines 295,Page 7. “Literature [62]”;

  • Clear the role of nanocomposites or nanofiber reinforcements in polymer matrices in bone regeneration

A7: Nano bone repair material is a kind of synthetic material with high mechanical strength, large specific surface area, good biocompatibility and biodegradability.  Therefore, biomimicry based on the properties and biochemical characteristics of nanomaterials is very important for promoting cell growth and inducing tissue regeneration.  Nanofibers can simulate the composition and structure of extracellular matrix and provide a good growth environment for cells. Nanofibers are ideal scaffolds for bone tissue engineering and regenerative medicine.

  • Check the citation style all over the document

A8: We have revised citation style all over the document.

  • Sec 4.1 adjust the line spacing

A9: We have revised the Section 4.1 in terms of line spacing on Page 17.

  • Table 3, it should be MiR-, check the table

A10: Thanks for your comment. We have modified Table 3.

  • Add a paragraph about How to prevent bone damage

A11: A paragraph on how to prevent bone injury is added in the conclusion and future perspectives on page 22, lines 623-640.

  • The review needs to be updated, most of the references are outdated, update to 2021, 2022

A12: We acknowledge the comments. We have revised and updated the reference list.

  • Check the outputs of all references, some lack page number or volume

A13: All references have been corrected in the reference list.  

  • Journal name abbreviated

A14: Thank you for your suggestion. All references have been revised and highlighted in bright yellow.

Round 2

Reviewer 2 Report

The manuscript can be accepted in the present form

This manuscript is a resubmission of an earlier submission. The following is a list of the peer review reports and author responses from that submission.